# Nomograms for Predicting Survival Outcomes in Patients with Neuroendocrine Neoplasms of the Gallbladder Undergoing Primary Tumor Resection: A Population-Based Study

**Yu-Rui Zhang †, Geng-Cheng Hu †, Meng-Ke Fan, Hai-Ling Yao, Chen Jiang, Hui-Ying Shi * and Rong Lin ***

Department of Gastroenterology, Union Hospital, Tongji Medical College,
Huazhong University of Science and Technology, Wuhan 430022, China
* Correspondence: shihuiying23@hotmail.com (H.-Y.S.); linrong@hust.edu.cn (R.L.)
† These authors contributed equally to this work.

**Abstract:** Background: Neuroendocrine neoplasms of the gallbladder (GB-NENs) are a rare group of histologically heterogeneous tumors, and surgical resection of the primary tumor is the mainstream treatment at the moment. The current study aimed to establish and validate novel nomograms for patients with GB-NENs undergoing primary tumor resection to predict the 6-, 12-, and 18-month overall survival (OS) and cancer-specific survival (CSS). Methods: Clinicopathological information of patients with GB-NENs undergoing primary tumor resection between 2004 and 2018 was derived from the Surveillance, Epidemiology, and End Results (SEER) database. Candidate prognostic factors were selected by Cox regression analyses, and the nomograms were constructed. Finally, concordance index (C-index), calibration plot, area under the curve from the receiver operating characteristic curve (AUC), and decision curve analysis (DCA) were utilized to assess the effective performance of the nomograms. Results: A total of 221 patients with GB-NENs undergoing resection were enrolled in this retrospective study. Using the Cox regression analyses, age, pathological classification, tumor size, and SEER stage were identified as the independent prognostic factors of patients with GB-NENs undergoing resection, and nomograms were constructed. The C-indexes of OS and CSS in training dataset were 0.802 (95% CI: 0.757–0.848) and 0.846 (95% CI: 0.798–0.895), while those of internal validation dataset were 0.862 (95% CI: 0.802–0.922) and 0.879 (95% CI: 0.824–0.934), respectively. Conclusions: Taken together, the nomograms are accurate enough to predict the prognostic factors of GB-NEN patients undergoing resection, allowing for treatment decision-making and clinical monitoring for future clinical work.

**Keywords:** neuroendocrine neoplasms of the gallbladder; nomogram; Cox regression; prognosis; SEER database

## 1. Introduction

Neuroendocrine neoplasms (NENs) represent an uncommon group of heterogeneous tumors, which originate from the diffuse neuroendocrine system and occur in almost all tissues and organs of the human body [1,2]. With the improvement of early disease detection and stage migration, the incidence of NENs is steadily growing. As reported by the SEER database, the latest annual age-adjusted incidence rate of NENs increased by 6.4 times from 1.09 per 100,000 in 1973 to 6.98 per 100,000 in 2012. In the general population, the incidence of NENs depends on the specific anatomical location, with the gastrointestinal tract being the most common site of occurrence, followed by the bronchopulmonary system [3].

However, due to the lack of neuroendocrine cells in the mucosa, NENs are rarely detected in the gallbladder, and neuroendocrine neoplasms of the gallbladder (GB-NENs) currently account for only 0.5% of all NEN cases [4,5]. Given the rarity of GB-NENs, we lacked sufficient understanding of the possible pathological mechanism driving the malignant transformation of GB-NENs. As a result, the only acceptable therapy for GB-NENs is the surgical removal of

the whole gallbladder [6]. However, because of the diverse biological behaviors and clinical characteristics of GB-NENs, prognostic prediction is a highly challenging task [1]. Lee et al. found the complete resection and application of postoperative adjuvant therapy may lead to a better clinical outcome for patients with neuroendocrine carcinomas of the gallbladder [7]. A study of 754 patients with GB-NENs selected from the National Cancer Database (NCDB) showed that advanced age, positive surgical margins, and large cell histology were significantly associated with shorter survival after resection. The prognosis of primary GB-NENs is worse than NENs of other gastrointestinal sites [8]. Notably, none of these risk factors can adequately predict the survival probability of each patient. There was an urgent need for a means of estimating the individualized prognosis of patients with GB-NENs undergoing resection through a large-scale cohort study.

Nomograms are generally considered to be a reliable and visualized statistical prediction model to accurately stratify risk by including important prognostic indicators of the diseases [9]. For many cancers, the use of nomograms is superior to the traditional staging systems; thus, it has been recommended as an alternative or even as a new standard [10]. In this study, we utilized the data from the SEER database to determine the significant prognostic indicators influencing OS and CSS of patients, and then the individualized nomograms were constructed. Furthermore, we used the validation dataset from the SEER database and our hospital to evaluate the clinical predictive performance of the nomograms.

## 2. Materials and Methods

### 2.1. Ethics Approval and Informed Consent

Since the SEER database is publicly accessible and all patient-identifiable information is de-identified, institutional review board approval and informed consent were not necessary for its usage. This study was a retrospective study and was approved by the Ethics Committee of Tongji Medical College, Huazhong University of Science and Technology (IORG number: IORG0003571). This research was carried out in compliance with the Helsinki Declaration of 1964 and its subsequent modifications' ethical standards.

### 2.2. Source Data and Screening Criteria

The SEER database provides demographic and clinicopathological information on cancer cases for approximately 30.0% of the United States population [11]. We collected GB-NEN cases from 18 population-based registries (2000–2018) in the SEER database using SEER* Stat 8.3.9 software. The strategy to identify GB-NEN cases is shown in Figure 1. Data of interest were extracted, including baseline demographics (age, sex, race, marital status, vital survival status, survival time), tumor features (pathological classification, tumor size, grade, SEER stage, lymph node (LN) metastasis, tumor metastasis), and treatment methods (other site surgery, lymph node surgery, radiotherapy, chemotherapy). The original data from the SEER database were examined for demographic and clinicopathological information. Patients who were divorced, separated, single, or widowed were categorized in the unmarried group for contrast with the married group due to the identical survival disadvantages associated with being unmarried. Patients were separated into two groups based on their age: those under 65 and those over 65. On the basis of the 2019 WHO classification of NEN [12], carcinoid tumors and atypical carcinoid tumors were classified as neuroendocrine tumors (NET) and neuroendocrine carcinoma; small-cell carcinoma and large-cell neuroendocrine carcinoma were classified as neuroendocrine carcinoma (NEC); and mixed adenoneuroendocrine carcinoma was classified as neuroendocrine–non-neuroendocrine neoplasm (MiNEN). Staging was carried out using the SEER Summary Stage 2000 (localized, regional, and distant). LN metastatic status was classified as non-metastatic, metastatic, or unknown status (N0, N1, or unknown). Tumor metastatic status was classified as non-metastatic, metastatic, or unknown status (M0, M1, or unknown). In this retrospective study, the primary endpoint was OS and CSS. The OS was the time from diagnosis to death of any cause. CSS was the time from diagnosis to death due to GB-NENs rather than any other cause of death.

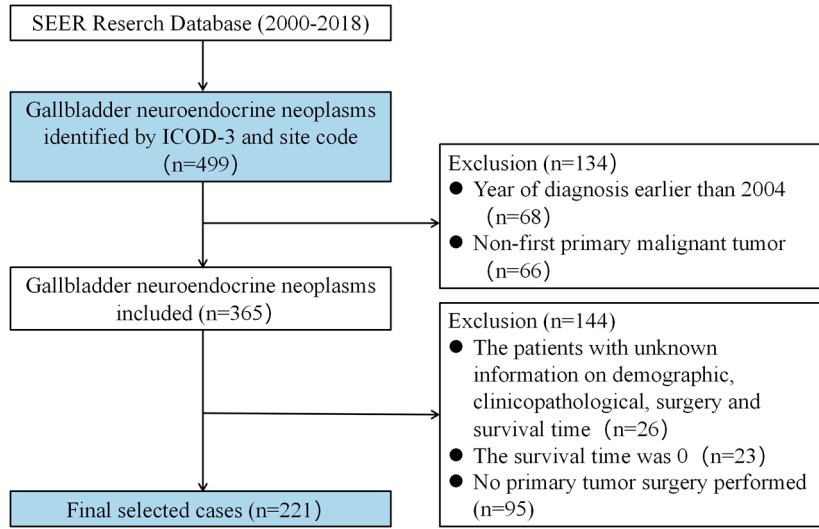

**Figure 1.** Flow diagram of the GB-NEN case selection procedure in the SEER database.

The inclusion criteria were defined as follows: (1) year of diagnosis ranged from 2004 to 2018, (2) primary site of the tumor was gallbladder (primary site code: C23.9), (3) histological types were confined to 8013/3, 8041/3, 8240/3, 8244/3, 8246/3, and 8249/3 according to International Classification of Disease for Oncology, third edition (ICDO-3). The exclusion criteria are described below: (1) unknown demographic information, (2) incomplete pathological diagnosis, (3) not first tumor, (4) no surgery on the primary site, and (5) survival time was zero or unknown.

The clinical records of 12 patients with GB-NENs receiving primary tumor surgery in the Wuhan Union Hospital were collected from 2011 to 2021. The clinical records and follow-up data were complete. The last follow-up was on November 15, 2021. The cause of the patient's death was specifically GB-NENs. The exclusion criteria were previously mentioned.

### 2.3. Construction and Validation of the Nomogram Models

The inclusion criteria were satisfied by 221 patients from the SEER database, who were then randomly split into a training set (N = 156) and an internal validation set (N = 65) in a 7:3 ratio. The cases collected from our hospital were used for external validation. The training set was used to construct the nomograms, and the internal and external validation sets were used for validation. The chi-squared test was used to compare categorical variables between the two groups. We performed univariate Cox regression analysis to select variables for constructing the prediction model. Multivariate Cox proportional hazards (CPH) regression analysis was performed to estimate OS and CSS, then the nomograms were constructed.

Concordance index (C-index), calibration plot, area under the curve (AUC) from the receiver operating characteristic (ROC) curve, and decision curve analysis (DCA) were utilized to evaluate the CPH prediction model. The C-index ranging from 0 to 1 was used to assess the prediction performance of the nomograms. A C-index greater than 0.7 usually indicates that the models have a good discriminatory ability [10]. The value of AUC ranged from 0.5 to 1, which was positively correlated with predictive ability. Calibration plots at 6, 12, and 18 months were drawn to compare the predicted survival probability with the survival probability observed in our study, and the 45-degree calibration curve represents the actual result of the perfect model [13]. The constructed nomograms were formulated using the Cox regression method with the best predictive accuracy to personalize the estimated probability of survival. As an emerging method, DCA was used to evaluate the potential clinical value of the prediction models [14]. All analyses were performed using R software version 4.0.5. *p*-values of less than 0.05 were defined as statistically significant.

## 3. Results

### 3.1. Patient Characteristics

A total of 221 patients from the SEER database with GB-NENs undergoing resection were enrolled. All patients from the SEER database were randomly assigned to two datasets in a 7:3 ratio: training (N = 156) and internal validation (N = 65). Of the patients included in this study, the majority of the patients are white (79.2%), and the percentage of females (64.7%) was higher than the percentage of males. Based on the 2019 WHO classification, NECs (57.0%) accounted for a greater percentage than NETs (38.9%) and MiNENs (4.1%). Among all patients, the median survival time was 22 months, and the mean tumor size was 27.07 ± 4.31 mm. There were 150 (67.9%) patients in stage N0 and 161 (72.9%) patients in stage M0. There were 147 (66.5%) patients without LN surgery and 195 (88.2%) patients without other site surgery. There were 73 (33.0%) patients who received chemotherapy and 30 (13.6%) patients who received radiotherapy. In addition, this research included 12 patients with GB-NENs who received surgical resection in our hospital. Of these, all patients were NECs, with a male to female ratio of 1:1, and 2/3 of patients had tumors of 2–5 cm in size. The demographic clinicopathological information of the patients is displayed in Table 1.

**Table 1.** Demographic and clinicopathological characteristics of patients with GB-NENs receiving primary tumor surgery.

| Variables | SEER Database (*n* = 221) | Training Dataset (*n* = 156) | Internal Validation Dataset (*n* = 65) | External Validation Dataset (*n* = 12) |
|---|---|---|---|---|
| Gender | | | | |
| Female | 143 (64.7) | 102 (65.4) | 41 (63.1) | 6 (50.0) |
| Male | 78 (35.3) | 54 (34.6) | 24 (36.9) | 6 (50.0) |
| Race | | | | |
| White | 175 (79.2) | 121 (77.6) | 54 (83.1) | 0 (0) |
| Black | 30 (13.6) | 21 (13.4) | 9 (13.8) | 0 (0) |
| Other | 16 (7.2) | 14 (9.0) | 2 (3.1) | 12 (100.0) |
| Age at diagnosis | | | | |
| ≤65 years | 119 (53.8) | 87 (55.8) | 32 (49.2) | 7 (58.3) |
| >65 years | 102 (46.2) | 69 (44.2) | 33 (50.8) | 5 (41.7) |
| Marital status | | | | |
| Married | 146 (66.1) | 97 (62.2) | 49 (75.4) | 12 (100.0) |
| Unmarried | 75 (33.9) | 59 (37.8) | 16 (24.6) | 0 (0) |
| Pathological classification | | | | |
| NET | 86 (38.9) | 61 (39.1) | 25 (38.5) | 0 (0) |
| NEC | 126 (57.0) | 89 (57.1) | 37 (56.9) | 12 (100.0) |
| MiNEN | 9 (4.1) | 6 (3.8) | 3 (4.6) | 0 (0) |
| N stage | | | | |
| N0 | 150 (67.9) | 110 (70.5) | 40 (61.5) | 7 (58.3) |
| N1 | 49 (22.2) | 32 (20.5) | 17 (26.2) | 5 (41.7) |
| Unknown | 22 (10.0) | 14 (9.0) | 8 (12.3) | 0 (0) |
| M Stage | | | | |
| M0 | 161 (72.9) | 115 (73.7) | 46 (70.8) | 6 (50.0) |
| M1 | 42 (19.0) | 29 (18.6) | 13 (20.0) | 6 (50.0) |
| Unknown | 18 (8.1) | 12 (7.7) | 6 (9.2) | 0 (0) |
| Tumor size | | | | |
| ≤2 cm | 92 (41.6) | 66 (42.3) | 26 (40.0) | 2 (16.7) |
| 2–5 cm | 50 (22.6) | 33 (21.2) | 17 (26.1) | 8 (66.6) |
| ≥5 cm | 32 (14.5) | 20 (12.8) | 12 (18.5) | 2 (16.7) |
| Unknown | 47 (21.3) | 37 (23.7) | 10 (15.4) | 0 (0) |
| Stage | | | | |
| Localized | 95 (43.0) | 72 (46.2) | 23 (35.4) | 0 (0) |
| Regional | 72 (32.6) | 48 (30.8) | 24 (36.9) | 6 (50.0) |
| Distant | 54 (24.4) | 36 (23.0) | 18 (27.7) | 6 (50.0) |
| Surgery at other sites | | | | |
| No | 195 (88.2) | 134 (85.9) | 61 (93.8) | 12 (100.0) |
| Yes | 26 (11.8) | 22 (14.1) | 4 (6.2) | 0 (0) |
| LN surgery | | | | |
| No | 147 (66.5) | 102 (65.4) | 45 (69.2) | 10 (83.3) |
| Yes | 74 (33.5) | 54 (34.6) | 20 (30.8) | 2 (16.7) |

**Table 1.** *Cont.*

| Variables | SEER Database (*n* = 221) | Training Dataset (*n* = 156) | Internal Validation Dataset (*n* = 65) | External Validation Dataset (*n* = 12) |
|---|---|---|---|---|
| Chemotherapy | | | | |
| No | 148 (67.0) | 103 (66.0) | 45 (69.2) | 7 (58.3) |
| Yes | 73 (33.0) | 53 (34.0) | 20 (30.8) | 5 (41.7) |
| Radiotherapy | | | | |
| No | 191 (86.4) | 135 (86.5) | 56 (86.2) | 11 (91.7) |
| Yes | 30 (13.6) | 21 (13.5) | 9 (13.8) | 1 (8.3) |

### 3.2. Identification of Prognostic Factors

As shown in Table 2, the univariate and multivariate Cox regression analysis was conducted to identify the prognostic factors in the training set. In the OS analysis, all 13 characteristics were reduced to seven potential predictors (age, pathological classification, N stage, M stage, SEER stage, tumor size, and chemotherapy) with $p < 0.05$. Then, the selected seven variables were incorporated into the multivariate Cox regression analysis. The CSS was analyzed in the same way, and five characteristics (gender, race, marital status, surgery at other sites, and LN surgery) were excluded. The selected eight variables were also incorporated in the multivariate Cox regression analysis. We considered the following characteristics to be important risk factors for OS decline: age greater than 65, classification of MiNENs, distant stage, and tumor size greater than 5 cm. Similar risk factors for CSS decline were identified. Additionally, tumor size greater than 2 cm and classification of NECs were linked to reduced OS and CSS, respectively.

**Table 2.** Univariate and multivariate Cox regression analyses of OS and CSS in training set.

| Variables | OS Univariate Analysis HR (95% CI) | *p*-Value | Multivariate Analysis HR (95% CI) | *p*-Value | CSS Univariate Analysis HR (95% CI) | *p*-Value | Multivariate Analysis HR (95% CI) | *p*-Value |
|---|---|---|---|---|---|---|---|---|
| Gender | | | | | | | | |
| Female | Ref. | | - | | Ref. | | - | |
| Male | 1.26 (0.80, 2.01) | 0.321 | | | 0.95 (0.54, 1.66) | 0.857 | | |
| Race | | | | | | | | |
| White | Ref. | | - | | Ref. | | - | |
| Black | 0.93 (0.46, 1.89) | 0.849 | | | 0.85 (0.36, 2.01) | 0.710 | | |
| Others | 1.48 (0.73, 3.01) | 0.272 | | | 1.41 (0.64, 3.14) | 0.396 | | |
| Age at diagnosis | | | | | | | | |
| ≤65 years | Ref. | | | | Ref. | | | |
| >65 years | 2.77 (1.74, 4.42) | <0.001 | 2.49 (1.42, 4.36) | 0.001 | 2.34 (1.37, 3.99) | 0.002 | 2.32 (1.16, 4.62) | 0.017 |
| Marital status | | | | | | | | |
| Married | Ref. | | - | | Ref. | | | |
| Unmarried | 0.783 (0.49, 1.26) | 0.313 | | | 0.60 (0.33, 1.07) | 0.082 | - | |
| Pathological classification | | | | | | | | |
| NET | Ref. | | | | Ref. | | | |
| NEC | 6.26 (3.34, 11.70) | <0.001 | 1.78 (0.74, 4.31) | 0.200 | 19.1 (5.93, 61.50) | <0.001 | 5.30 (1.33, 21.21) | 0.018 |
| MiNEN | 12.60 (4.33, 36.40) | <0.001 | 5.57 (1.52, 20.46) | 0.010 | 33.9 (7.49, 153) | <0.001 | 15.44 (2.71, 87.94) | 0.002 |
| N stage | | | | | | | | |
| N0 | Ref. | | | | Ref. | | | |
| N1 | 1.78 (1.03, 3.09) | 0.038 | 0.65 (0.34, 1.24) | 0.194 | 2.06 (1.14, 3.75) | 0.017 | 0.67 (0.33, 1.35) | 0.266 |
| Unknown | 1.19 (0.47, 3.04) | 0.710 | 2.31 (0.56, 9.58) | 0.248 | 1.50 (0.58, 3.87) | 0.402 | 2.87 (0.66, 12.56) | 0.162 |
| M stage | | | | | | | | |
| M0 | Ref. | | | | Ref. | | | |
| M1 | 5.64 (3.34, 9.53) | <0.001 | 1.91 (0.66, 5.47) | 0.231 | 6.45 (3.62, 11.5) | <0.001 | 1.29 (0.42, 3.96) | 0.655 |
| Unknown | 0.99 (0.30, 3.23) | 0.984 | 0.31 (0.05, 1.95) | 0.212 | 1.24 (0.37, 4.11) | 0.726 | 0.25 (0.04, 1.76) | 0.163 |
| Tumor size | | | | | | | | |
| ≤2 cm | Ref. | | | | Ref. | | | |
| 2–5 cm | 3.74 (1.95, 7.18) | <0.001 | 1.75 (0.81, 3.79) | 0.156 | 8.30 (3.26, 21.10) | <0.001 | 3.29 (1.13, 9.53) | 0.028 |
| ≥5 cm | 5.45 (2.69, 11.0) | <0.001 | 2.88 (1.31, 6.33) | 0.009 | 12.6 (4.73, 33.50) | <0.001 | 5.55 (1.92, 16.04) | 0.002 |
| Unknown | 4.28 (2.29, 7.99) | <0.001 | 4.15 (2.07, 8.30) | <0.001 | 9.50 (3.81, 23.70) | <0.001 | 9.04 (3.19, 25.64) | <0.001 |
| SEER Stage | | | | | | | | |
| Localized | Ref. | | | | Ref. | | | |
| Regional | 4.01 (2.20, 7.32) | <0.001 | 2.50 (1.09, 5.72) | 0.030 | 5.92 (2.62, 13.40) | <0.001 | 2.17 (0.78, 6.00) | 0.136 |
| Distant | 10.10 (5.43, 18.90) | <0.001 | 4.50 (1.39, 14.62) | 0.012 | 17.00 (7.52, 38.50) | <0.001 | 6.18 (1.52, 25.08) | 0.011 |
| Surgery at other sites | | | | | | | | |
| No | Ref. | | - | | Ref. | | - | |
| Yes | 1.56 (0.85, 2.84) | 0.149 | | | 1.69 (0.87, 3.27) | 0.120 | | |
| LN surgery | | | | | | | | |
| No | Ref. | | - | | Ref. | | - | |
| Yes | 0.89 (0.55, 1.43) | 0.620 | | | 1.17 (0.69, 1.99) | 0.564 | | |
| Chemotherapy | | | | | | | | |
| No | Ref. | | | | Ref. | | | |
| Yes | 2.41 (1.52, 3.80) | <0.001 | 0.96 (0.52, 1.76) | 0.900 | 3.20 (1.89, 5.43) | <0.001 | 0.96 (0.47, 1.96) | 0.918 |
| Radiotherapy | | | | | | | | |
| No | Ref. | | - | | Ref. | | | |
| Yes | 1.68 (0.94, 3.02) | 0.079 | | | 2.00 (1.05, 3.80) | 0.034 | 0.92 (0.44, 1.89) | 0.812 |

### 3.3. Construction and Validation of the Nomograms

Age, pathological classification, tumor size, and SEER stage were identified as independent prognostic factors by Cox regression analysis (Table 2). Then, the prediction

models that contained the above predictors were constructed using the CPH regression model based on the training set and are shown as the nomograms (Figure 2). In internal validation, the C-indexes for the nomograms of OS and CSS were 0.802 (95% confidence interval (CI): 0.757–0.848) and 0.846 (95% CI: 0.798–0.895) in the training set, respectively. In the internal validation set, the C-indexes for the models of OS and CSS were 0.862 (95% CI: 0.802–0.922) and 0.879 (95% CI: 0.824–0.934), respectively, presenting a good discriminatory ability of the nomograms. In addition, AUC values at three different time points (6, 12, and 18 months) were calculated to evaluate the predictive accuracy of the prediction models. In OS analysis, for each of the three time points, the AUC values of the training set were 0.772, 0.819, and 0.892; the AUC values of the internal validation set were 0.951, 0.878, and 0.913 (Figure 3A,B). In CSS analysis, for each of the three time points, the AUC values of the training set were 0.791, 0.848, and 0.916; the AUC values of the internal validation set were 0.951, 0.889, and 0.922 (Figure 3C,D). Furthermore, the calibration plots showed that the nomograms had good performance in predicting the 12-month OS and CSS in the training and internal validation sets (Figures 4 and 5).

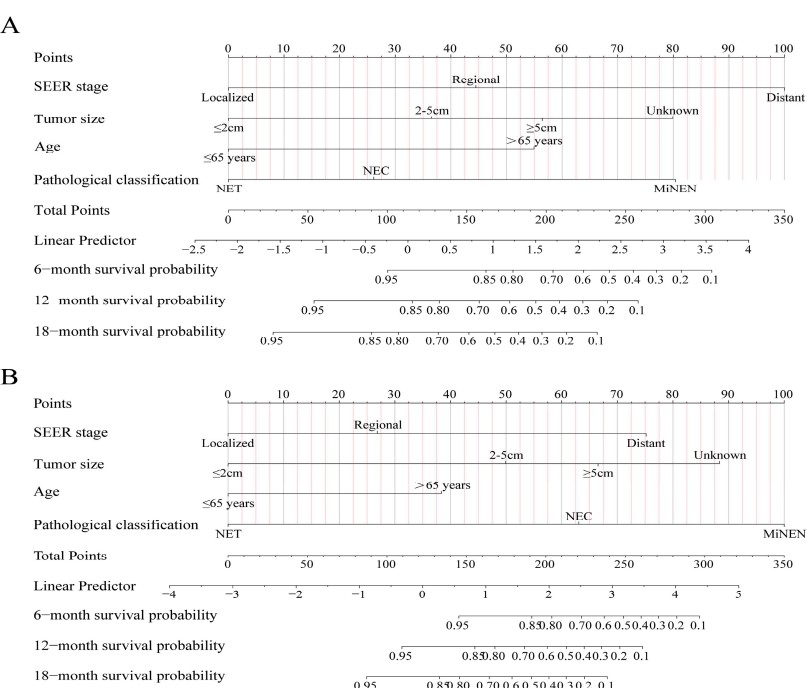

**Figure 2.** Predictive nomograms. (**A**) Nomogram for the prediction of 6-, 12-, and 18-month overall survival. (**B**) Nomogram for the prediction of 6-, 12-, and 18-month cancer-specific survival.

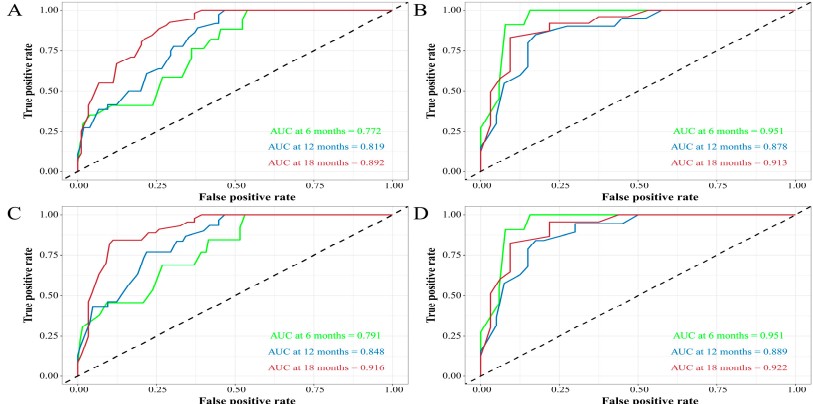

**Figure 3.** ROC curves for predicting 6-, 12-, and 18-month overall survival (OS) and cancer-specific survival (CSS) rates of the nomograms. (**A,B**) ROC curves for OS in training and internal validation sets. (**C,D**) ROC curves for CSS in training and internal validation sets.

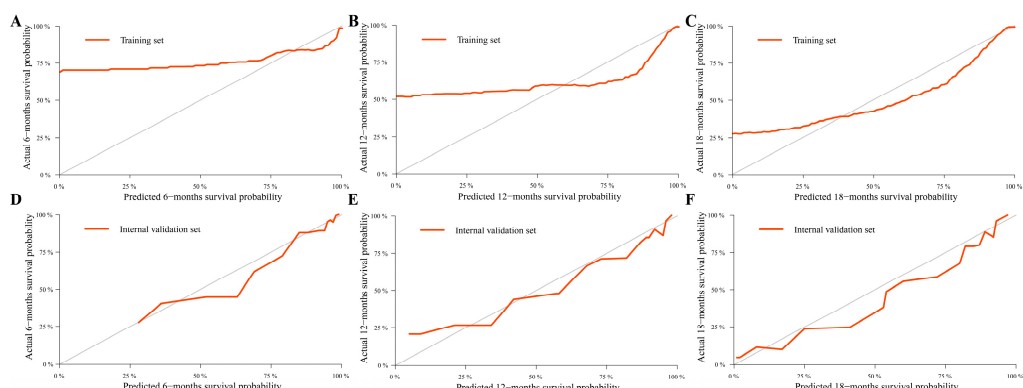

**Figure 4.** Calibration plots for predicting overall survival (OS). Calibration curves for predicting 6-, 12-, and 18-month OS in the training set (**A**–**C**) and internal validation set (**D**–**F**). The black line represents the comparison between the observed survival probability, and the prediction probability the gray line represents the ideal prediction.

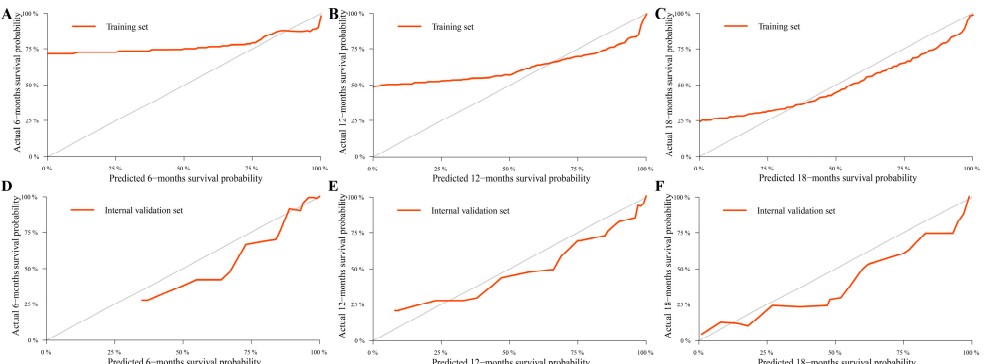

**Figure 5.** Calibration plots for predicting cancer-specific survival (CSS). Calibration curves for predicting 6-, 12-, and 18-month CSS in the training set (**A**–**C**) and internal validation set (**D**–**F**).

### 3.4. Clinical Application of the Nomograms

In the DCA curves, when the threshold probability was greater than 25%, the net benefits of the constructed nomograms and the SEER stage system were comparable in predicting the OS and CSS. Within this range, the clinical performance of the nomograms was significantly better than the SEER stage system in predicting OS and CSS (Figure 6).

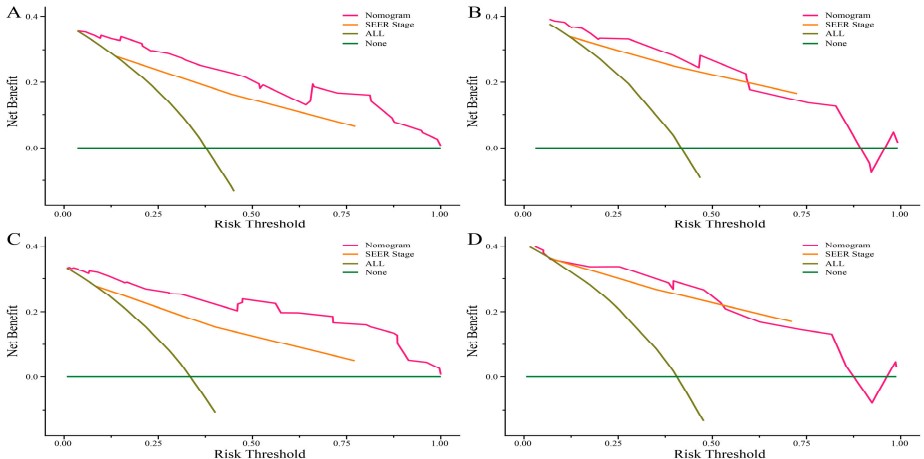

**Figure 6.** Decision curve analysis (DCA) of the nomogram and SEER stage system for the overall survival (OS) and cancer-specific survival (CSS) prediction of patients with neuroendocrine neoplasms of the gallbladder undergoing primary tumor resection. (**A**) The training and (**B**) internal validation sets for the OS. (**C**) The training and (**D**) internal validation sets for the CSS.

In external validation, we analyzed the data of GB-NEN patients who received primary tumor surgery (n = 12) at Wuhan Union hospital from 2010 to 2021. The last follow-up was on 15 November 2021. Demographics and clinicopathological characteristics are shown in Table 1. Given the rarity of GB-NENs, our data of GB-NENs can only validate CSS at 6 and 12 months. In CSS analysis, the C-indexes were 0.777 (95% CI: 0.608–0.946). In addition, the AUCs of 6- and 12-month (0.972 and 0.861) nomograms were greater than the SEER stage system (0.833 and 0.833) (Figure 7). The results suggested that our prediction model had good applicability and predictive performance in external data.

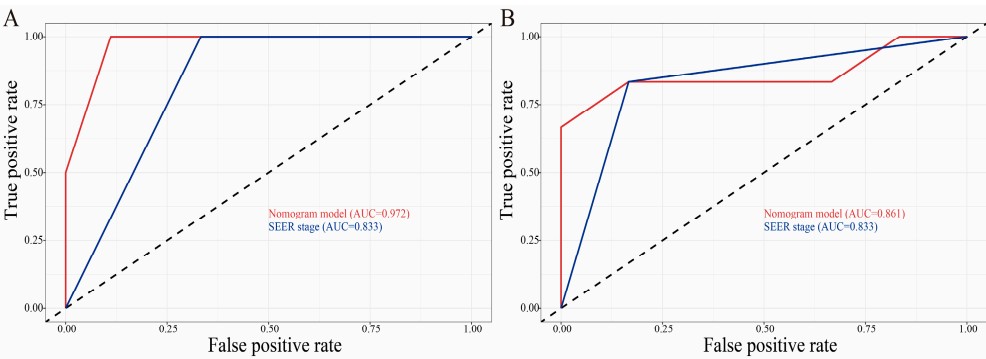

**Figure 7.** External validation of nomogram compared with SEER stage system in 12 GB-NEN cases from Wuhan Union hospital. ROC curves of the nomogram and SEER stage system in the prediction of prognosis at 6 and 12 months (**A**,**B**).

## 4. Discussion

Neuroendocrine neoplasms are highly heterogeneous diseases, and the original site of the neoplasm affects its prognosis. As shown in the previous study [15], the prognoses of neuroendocrine neoplasms of the gallbladder, bile duct, and ampulla of Vater (AoV) are different. GB-NENs are relatively rare tumors, and the main treatment is surgical resection of the whole gallbladder. However, the prognostic factors of patients undergoing primary tumor resection are still ambiguous. Thus, it is very important and necessary to carry out an independent study on the prognosis of patients with GB-NENs undergoing primary tumor resection. In the current study, we constructed individualized nomograms to predict the prognosis of patients. After bootstrap validation internally, the CPH survival models were evaluated by C-indexes, AUC, calibration plots, and DCAs, showing the nomograms have good discriminatory capabilities and calibration. Thirteen clinical characteristics were collected as potential prognostic factors, and these most valuable variables were selected by COX regression analysis and incorporated into the nomograms. Age, pathological classification, tumor size, and SEER stage were identified as significant variables to measure the prognostic score along nomogram scales. These characteristics are easily obtained from patients, making it easy to use the models in real life.

Age as a personal characteristic is widely considered to be a prognostic predictor for gallbladder cancer as well as neuroendocrine neoplasms in other sites, such as gastric NENs, colorectal NENs, and pancreatic NENs [16–20]. With age, the chance of cancer-causing genetic mutations increases. Aging is linked to highly reproducible DNA methylation alterations, which may explain why older people are more likely to get cancer [21]. Older patients with GB-NENs often have a worse prognosis than younger patients [22]. In an NCDB-based study, 300 patients who underwent resection of GB-NENs were analyzed, and the results suggested that elderly patients may have worse survival [8]. The poorer survival rate of elderly GB-NEN patients is due to coexisting diseases and weakened resistance to injury caused by tumor invasion, surgery, or adjuvant therapy [16,17]. Therefore, age can be considered an important prognostic factor for GB-NEN patients.

In addition to personal characteristics, tumor biological characteristics may play an important role in the prognosis of GB-NEN patients. In this study, NEC and MiNEN types have a detrimental effect on the CSS of patients compared with NET types. There are

numerous pathological types in GB-NENs, and each tumor may behave very distinctively with regard to local invasion and metastasis. In general, NETs are well differentiated, with modest mitotic activity and invasiveness, while NEC has a highly aggressive growth tendency and is prone to developing distant metastases early [6]. Moreover, the overall survival for gastrointestinal MiNEN is significantly shorter than that of NEC. Larger tumor size and advanced stage may indicate poorer survival of GB-NEN patients. The high stage means extensive tumor infiltration, reflecting the high malignancy and rapid progression of GB-NENs, which often leads to the involvement of adjacent important organs [23]. Furthermore, the expansion of the tumor infiltration area will make radical surgery difficult, and the probability of tumor recurrence after surgery will be greatly increased [24]. Our study further verified that pathological type, tumor size, and SEER stage were the significant risk factors for GB-NEN patients receiving primary tumor surgery, which represented the inherent characteristics of the tumor affecting prognosis. The GB-NENs are currently staged according to the same AJCC staging criteria as gallbladder adenocarcinoma, but the AJCC staging system may not be very suitable for neuroendocrine neoplasms [25,26]. The prediction models seem to be a very practical and effective tool, especially for GB-NEN patients receiving primary tumor surgery.

At present, surgery remains the cornerstone of treatment for localized tumors, and the systemic treatment choices for patients with advanced NENs have expanded considerably [1]. However, due to the lack of sufficient data, the treatment strategy for patients with GB-NENs is still controversial. Iype et al. reported that chemotherapy drugs, including cisplatin, carboplatin, and etoposide, perhaps lead to partial response and added a marginal advantage for patients with GB-NENs [27]. Similarly, Chorath et al. reported a patient with high-grade gallbladder neuroendocrine carcinomas experiencing partial response to carboplatin, etoposide, nivolumab, and ipilimumab [28]. The role of radiotherapy in the treatment of GB-NENs is unclear since NENs are generally insensitive to traditional radiotherapy [6]. In our analysis, chemotherapy and radiotherapy were not identified as potential prognostic factors in patients with GB-NENs. However, the results of univariate Cox regression analysis showed that chemotherapy had a negative effect on OS and CSS, while radiotherapy only had a positive effect on CSS. Thus, in the treatment of GB-NENs, chemotherapy may not be recommended for a routine postoperative adjuvant therapy, and radiotherapy was recommended when the condition permitted. Despite this, it is still important and necessary to clarify the efficacy of chemotherapy and radiotherapy in larger sample studies.

We developed and validated the individual nomograms for OS and CSS in patients with GB-NENs. This study has some advantages. First, to avoid heterogeneity between different medical institutions, a large sample dataset from the SEER database was combined with the sample dataset from a single medical institution. Second, the variables incorporated in the nomograms are available and are often easily obtained in daily clinical practice. We also analyzed the effectiveness of chemotherapy and radiotherapy in the treatment of GB-NENs. The results showed that chemotherapy may negatively affect the prognosis of GB-NENs, and radiotherapy was recommended when necessary. Third, the nomograms had good calibration and discriminatory ability (the C-indexes for OS and CSS are 0.802 and 0.846 respectively). This can help us make better decisions in the actual clinical environment. Meanwhile, we plotted the DCA curves of the clinical impact of the nomograms, and the results indicated that our nomograms had greater clinical prediction performance than the SEER stage system. The limitation of this study lies in the retrospective nature with potential selective bias, which may not reflect problems encountered in actual clinical practice. In addition, our nomograms were constructed using only four clinicopathological characteristics, lacking other important variables, such as Ki-67, chromogranin A, and neuron-specific enolase [29,30]. Regrettably, these characteristics are not included in the SEER database; thus, further studies should incorporate Ki-67, chromogranin A, and neuron specific enolase for analysis. Although the C-indexes, AUC, calibration plots, and

DCAs were applied to validate the nomograms, multi-center validation of large samples is still necessary. More work to enhance the validity of the prediction model is warranted.

## 5. Conclusions

The individualized nomograms to predict survival of GB-NEN patients undergoing primary tumor surgery were developed based on independent variables, including age, pathological classification, tumor size, and SEER stage. We consider the prognostic models practical and effective tools for clinicians that can facilitate prediction of prognosis in patients with GB-NENs after primary tumor surgery.

**Author Contributions:** Y.-R.Z. and G.-C.H. contributed equally to this work. Conceptualization: Y.-R.Z., G.-C.H., H.-Y.S. and R.L.; methodology: Y.-R.Z., G.-C.H., H.-Y.S. and R.L.; software: Y.-R.Z., H.-L.Y. and M.-K.F.; data curation: G.-C.H., H.-L.Y. and C.J.; data analysis: M.-K.F., H.-L.Y. and C.J.; writing—original draft: Y.-R.Z. and G.-C.H.; writing—review and editing: H.-Y.S. and R.L. All authors have read and agreed to the published version of the manuscript.

**Funding:** This study was supported by the National Natural Science Foundation of China (funder: Rong Lin, funding numbers: 81770539 and 81974068), the National key research and development program of China (funder: Rong Lin, funding number: 2017YFC0110003), and the Interdisciplinary Program of Wuhan National High Magnetic Field Center (funder: Rong Lin, funding number: WHMFC 202104), Huazhong University of Science and Technology.

**Institutional Review Board Statement:** This retrospective study was approved by the Ethics Committee of Tongji Medical College, Huazhong University of Science and Technology (IORG number: IORG0003571), and the final approval date was 11 October 2020.

**Informed Consent Statement:** Informed consent was obtained from each patient, and the study conforms to the provisions of the Declaration of Helsinki.

**Data Availability Statement:** The authors declare that the research was conducted in the absence of any commercial or financial relationships that could be construed as a potential conflict of interest.

**Conflicts of Interest:** All authors certify that they have no affiliations with or involvement in any organization or entity with any financial interest or non-financial interest in the subject matter or materials discussed in this manuscript.

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
