# Peer review of "Nomograms for Predicting Survival Outcomes in Patients with Neuroendocrine Neoplasms of the Gallbladder Undergoing Primary Tumor Resection: A Population-Based Study"

_curroncol, doi:10.3390/curroncol30030221_

Round 1
Reviewer 1 Report (Previous Reviewer 1)
Comments on the revised manuscript:
Nomograms for predicting survival outcomes in patients with neuroendocrine neoplasms of the gallbladder undergoing primary tumor resection: a population-based study
The manuscript has been revised by the authors according to reviewers’ comments.
The main concern - the study is of retrospective design, with a low number of patients. Consequently, the quality and reliability of the presented evidence is low. However, considering the rarity of the disease, there is at least some evidence.
I do not have further comments.
Reviewer 2 Report (Previous Reviewer 2)
thank you very much for your email. In my opinion, the paper is now far better than in the previous form, so it is a pleasure for me to communicate that it can be published in this form. Best regardsThis manuscript is a resubmission of an earlier submission. The following is a list of the peer review reports and author responses from that submission.
Round 1
Reviewer 1 Report
Comments on:
Nomograms for predicting survival outcomes in patients with neuroendocrine neoplasms of the gallbladder undergoing primary tumor resection: a population-based study
The authors have done a great job to perform this study, trying to better understand the impact of neuroendocrine tumors on patient survival, aiming to personalize, at least partially, treatment and surveillance options. However, some comments emerged while reading the manuscript.
The study is of retrospective design, with a low number of patients, which are further split into three different cohorts… already per se indicating possibility of biases. The external validation cohort is also very small, consisting of 12 patients only. In addition, characteristics of these patients should be included in the body text as these are of importance.
It is of extreme importance to define the area of interest as the differences of morphological appearance, biological behavior, and clinical presentation between NET and NEC are very well described. NET may be indolent, while NEC – no. The histological types 8013/3, 8041/3, 8240/3, 8244/3, 8246/3, 8249/3 (ICDO-3) as listed in the inclusion criteria refer to carcinomas. However only 40% of included patients were diagnosed with NEC, while the remaining 60% were diagnosed with NET or “unknown”. Further detailing on patient and tumor characteristics lack data on Ki-67, a proliferation marker for human tumor cells to better understand biological behavior of this very heterogeneous group of tumors. It is not clear what does the created nomogram show – prognosis for all NETs or specifically for NECs.
The statistical analysis seems to be valid and produces desired outcomes; however, there are some doubts as to how the variables were identified, how the data was interpreted. It is hard to perceive that NEC has no impact on survival, as well as regional or distant metastases, while SEER stages appear to be important. Not to speak about lacking and potentially strong variables, such as Ki67, Chromogranin A, neuron-specific enolase, as mentioned by the authors also. Statistical analysis is of the utmost importance; however, interpreting results logical approach has to be utilized. Favorable statistical data should not lead to illegitimate conclusions.
Conclusions include speculative part on individualized decisions and treatments as these statements are not based on the results of this study.
The tables included are of a very poor quality, hardly readable.
Author Response
Dear Editors and Reviewers:
Thank you for your letter and for the reviewers’ comments concerning our manuscript entitled “Nomograms for predicting survival outcomes in patients with neuroendocrine neoplasms of the gallbladder undergoing primary tumor resection: a population-based study” (Manuscript ID: curroncol-2069019). Those comments are all valuable and very helpful for revising and improving our paper, as well as the important guiding significance to our researches. We have studied comments carefully and have made correction which we hope meet with approval. Revised portion are marked in red in the paper. The main correction in the paper and the responds to the reviewer’s comments are as follows:
Response to Reviewer 1 Comments
- Point 1: The study is of retrospective design, with a low number of patients, which are further split into three different cohorts... already per se indicating possibility of biases. The external validation cohort is also very small, consisting of 12 patients only. In addition, characteristics of these patients should be included in the body text as these are of importance.
Response 1: Thank you very much for your suggestion. Unfortunately, due to the rarity of neuroendocrine neoplasms of the gallbladder, this is currently the largest number of cases available in the public database and in our hospital database. In addition, according to the reviewer’s suggestion, we have placed the optimized table with external validation data in Table 1 of the body text and described it in lines 138-140.
- Point 2: It is of extreme importance to define the area of interest as the differences of morphological appearance, biological behavior, and clinical presentation between NET and NEC are very well described. NET may be indolent, while NEC - no. The histological types 8013/3, 8041/3, 8240/3, 8244/3, 8246/3, 8249/3 (ICDO-3) as listed in the inclusion criteria refer to carcinomas. However only 40% of included patients were diagnosed with NEC, while the remaining 60% were diagnosed with NET or “unknown”. Further detailing on patient and tumor characteristics lack data on Ki-67, a proliferation marker for human tumor cells to better understand biological behavior of this very heterogeneous group of tumors. It is not clear what does the created nomogram show - prognosis for all NETs or specifically for NECs.
Response 2: Thank you for your professional suggestions. Our previous classification criteria of neuroendocrine neoplasms of the gallbladder (GB-NENs) was mainly based on the 2019 WHO classification of tumours of the digestive system (PMID: 31433515), and was differentiated according to the index “grade” in SEER database. The classification criteria based the index “grade” did not distinguish NET, NEC and MiNEN well due to the differences between index “grade” in SEER database and index “grade” in WHO classification (in SEER database, G1-G4 was considered as well differentiated, moderately differentiated, poorly differentiated, and undifferentiated, but in WHO classification, G1-G3 was considered as well differentiated). Our current classification criteria is mainly based on the classification criteria of previous articles (PMID: 34350109, 31398710), the carcinoid tumors and atypical carcinoid tumors as gallbladder neuroendocrine tumors (NET), neuroendocrine carcinoma, small-cell carcinoma and large-cell neuroendocrine carcinoma as gallbladder neuroendocrine carcinoma (NEC), and mixed adenoneuroendocrine carcinoma as neuroendocrine–non-neuroendocrine neoplasm (MiNEN). The adjusted content and subsequent analysis results are presented in lines 79-91 and 241-246.
Ki67 index has been widely used as a proliferation marker and prognostic factor for gastroenteropancreatic neuroendocrine neoplasms (PMID: 29134440). Determining the cut-off value of Ki67 index that predict the response to treatment and prognosis of GB-NENs remains a problem with great clinical value. However, due to the lack of Ki67 index in the SEER database, we are unable to include it in the constructed prediction model, which is one of the limitations of our study. We sincerely hope that future studies with large samples will address this question.
Finally, after redefining the classification criteria of neuroendocrine tumors in this study, NET patients account for 38.9%, NEC patients account for 57%, and MiNEN patients account for 4.1%. In addition, pathological classification was identified as an important prognostic factor for GB-NENs. As shown in Figure 3, the adjusted prediction model has good prognostic ability for GB-NENs patients.
- Point 3: The statistical analysis seems to be valid and produces desired outcomes; however, there are some doubts as to how the variables were identified, how the data was interpreted. It is hard to perceive that NEC has no impact on survival, as well as regional or distant metastases, while SEER stages appear to be important. Not to speak about lacking and potentially strong variables, such as Ki67, Chromogranin A, neuron-specific enolase, as mentioned by the authors also. Statistical analysis is of the utmost importance; however, interpreting results logical approach has to be utilized. Favorable statistical data should not lead to illegitimate conclusions.
Response 3: Thank you for your comments. Based on previous study (PMID: 34350109, 31398710), we changed the original tumor size classification criteria (≤2cm, >2cm, Unknown) to a new criteria (≤2cm, 2-5cm, ≥5cm, Unknown), and changed the pathological classification criteria (NET, NEC, Unknown) to a new criteria (NET, NEC, MiNEN). Recalculated COX regression analysis results in Table 2 emphasized the impact of age, SEER stage, pathological classification, and tumor size on survival. Multivariate COX regression results suggest that classification of NEC and MiNEN are important risk factors for reducing cancer-specific survival (CSS), classification of MiNEN is important risk factor for reducing overall survival (OS), and classification of NEC has little impact on OS. For tumor size, tumor size greater than 2cm is an important risk factor for reduced CSS, while tumor size greater than 5cm is an important risk factor for reduced CSS. Moreover, older age and distant SEER stage are important risk factors for reduced OS and CSS. These results show that the clinicopathological characteristics of tumors have a significant impact on the survival of patients.
Ki67 index has been widely used as a proliferation marker and prognostic factor for gastroenteropancreatic neuroendocrine neoplasms (PMID: 29134440). Chromogranin A and neuron-specific enolase are potential tumor biomarkers for predicting the prognosis of NENs (PMID: 36233409). The SEER database lacks data on Ki67, chromogranin A, and neuron-specific enolase, so we could not analyze the relationship between these important indicators and the prognosis of GB-NENs in this study, which is the main deficiency of our study.
- Point 4: Conclusions include speculative part on individualized decisions and treatments as these statements are not based on the results of this study.
Response 4: Thank you very much for your suggestion. We have changed the content of this section and replaced it with “The individualized nomograms to predict survival of GB-NENs patients undergoing primary tumor surgery was developed based on independent variables, including age, pathological classification, tumor size, and SEER stage. We consider the prognostic models are practical and effective tools for clinicians that can facilitate prediction of prognosis in patients with GB-NENs after primary tumor surgery” in lines 294-298.
- Point 5: The tables included are of a very poor quality, hardly readable.
Response 5: Thank you very much for your suggestion. We have improved the quality of all tables and figures. The optimized diagram has been replaced in the main text.
Special thanks to you for your good comments!
Reviewer 2 Report
1)lines 9-10: “Neuroendocrine neoplasms of the gallbladder (GB-NENs) are a rare group of histologically heterogeneous tumors”. Please specify better what do you mean saying they are histologically heterogeneous.
2) lines 19-20 : “A total of 221 patients with GB-NENs undergoing resection were enrolled in our study.”, please specify they were retrospectively enrolled.
3) line 38 “the incidence of NENs depends on the specific anatomical location, with the highest incidence in the gastrointestinal pancreas and lungs.” The sentence has to be changed. In this form it can be misleading.
4) lines 69-70 “informed consent was not required in this study. The 69 patients from our hospital included in this study all provided oral consent.” This sentence must be better explained and argued on the basis of the current legislation. The current legislation regarding informed consent represents a protection for the Patients. If not adequately justified, the absence of informed consent represents a violation of the Patient's rights.
5) lines 101-103 “All cases from the SEER database were randomly assigned into two groups (7:3), 101 which consisted of the training set (n = 154, 70% of total cases) and the internal validation 102 set (n = 67, 30% of total cases)”. This statement should be better clarified by the Authors.
6) lines 129-130 “were women, and 146 129 (66.1%) were married”. Is this information relevant? Not in my opinion.
7) Lines 209-210: “Neuroendocrine neoplasms are highly heterogeneous diseases, and the prognosis depends on the original site.”. It is not completely true, probably the authors should add “also” to the sentence.
8)Lines 213-214. “However, the prognostic factors of patients undergoing primary tumor resection are still unknown”. Are the authors sure about this sentence? Please check into literature.
9) lines 282-283: “lacking other important variables, such as Ki-67, chromogranin A, and neuron-specific enolase[29, 30], which may reduce the prediction accuracy of our nomograms.”. Chromogranin and NSE are two well accepted and validated markers for NETs, the authors should improve this sentence taking it into consideration.
Author Response
Dear Editors and Reviewers:
Thank you for your letter and for the reviewers’ comments concerning our manuscript entitled “Nomograms for predicting survival outcomes in patients with neuroendocrine neoplasms of the gallbladder undergoing primary tumor resection: a population-based study” (Manuscript ID: curroncol-2069019). Those comments are all valuable and very helpful for revising and improving our paper, as well as the important guiding significance to our researches. We have studied comments carefully and have made correction which we hope meet with approval. Revised portion are marked in red in the paper. The main correction in the paper and the responds to the reviewer’s comments are as follows:
Response to Reviewer 2 Comments
- Point 1: lines 9-10: “Neuroendocrine neoplasms of the gallbladder (GB-NENs) are a rare group of histologically heterogeneous tumors”. Please specify better what do you mean saying they are histologically heterogeneous.
Response 1: Thank you very much for your comments. Previous literature reported that neuroendocrine neoplasms of gallbladder (GB-NENs) probably originate from either a multipotent stem cell or neuroendocrine cells in intestinal or gastric metaplasia of the gallbladder epithelium (PMID:26911175, 20375728). Thus, there are multiple pathological types of GB-NENs, including neuroendocrine tumor, neuroendocrine carcinoma, small cell carcinoma, and large cell neuroendocrine carcinoma, etc. To some extent, due to the different types of neuroendocrine cells of tumor origin, the individual behavior of each type of NENs may vary substantially in terms of local invasion and metastasis. Therefore, we consider that GB-NENs are histologically heterogeneous.
- Point 2: lines 19-20: “A total of 221 patients with GB-NENs undergoing resection were enrolled in our study.”, please specify they were retrospectively enrolled.
Response 2: Thank you very much for your comments. We have clearly indicated that this study is a retrospective study in line 20.
- Point 3: line 38 “the incidence of NENs depends on the specific anatomical location, with the highest incidence in the gastrointestinal pancreas and lungs.” The sentence has to be changed. In this form it can be misleading.
Response 3: Thank you very much for your suggestion. we have changed the content of the sentence by replacing it with “the incidence of NENs depends on the specific anatomical location, with the gastrointestinal tract being the most common site of occurrence, followed by the bronchopulmonary system” in lines 38-39.
- Point 4: lines 69-70 “informed consent was not required in this study. The 69 patients from our hospital included in this study all provided oral consent.” This sentence must be better explained and argued on the basis of the current legislation. The current legislation regarding informed consent represents a protection for the Patients. If not adequately justified, the absence of informed consent represents a violation of the Patient's rights.
Response 4: Thank you for your professional suggestions. We strongly agree with you. Unfortunately, our statements in this section may be misleading. Since SEER database is publicly available and all patient’s personal identifying information are de-identified, institutional review board approval and informed consent were not required for its usage. This study was a retrospective study and was approved by the Ethics Committee of Tongji Medical College, Huazhong University of Science and Technology (IORG number: IORG0003571). This study was performed in accordance with the 1964 Helsinki Declaration and its later amendments ethical standards. The corresponding changes are in lines 65-70.
- Point 5: lines 101-103 “All cases from the SEER database were randomly assigned into two groups (7:3), 101 which consisted of the training set (n = 154, 70% of total cases) and the internal validation 102 set (n = 67, 30% of total cases)”. This statement should be better clarified by the Authors.
Response 5: Your suggestion is greatly appreciated. We have changed the presentation of the sentence by replacing it with “The inclusion criteria were satisfied by 221 patients from the SEER database, who were then randomly split into training set (N = 156) and internal validation set (N = 65) in a 7:3 ratio.” in lines 107-108.
- Point 6: lines 129-130 “were women, and 146 129 (66.1%) were married”. Is this information relevant? Not in my opinion.
Response 6: Thank you for your suggestion. We have deleted the irrelevant information in lines 130-140.
- Point 7: Lines 209-210: “Neuroendocrine neoplasms are highly heterogeneous diseases, and the prognosis depends on the original site.”. It is not completely true, probably the authors should add “also” to the sentence.
Response 7: According to the reviewer’s suggestion, we have changed the presentation of the sentence by replacing it with “Neuroendocrine neoplasms are highly heterogeneous diseases, and its original site affects its prognosis.” in lines 215-216.
- Point 8: Lines 213-214. “However, the prognostic factors of patients undergoing primary tumor resection are still unknown”. Are the authors sure about this sentence? Please check into literature.
Response 8: Thank you very much for your suggestion. Our statement in this part is indeed controversial, and then we carefully reviewed the previous literature. In the literature published by Cen et al., they reported that age, marital status, tumor size and SEER stage were predictive factors for the survival of patients with neuroendocrine neoplasms of gallbladder (GB-NENs) (PMID: 31398710). Gogna et al. reported similar results (PMID: 34236488). However, Zhou et al. reported that age, tumor size, pathological classification, SEER stage and surgery are important predictors of biliary neuroendocrine neoplasms (PMID: 34350109). Their conclusions about the prognostic factors of the survival in GB-NENs patients are controversial, especially for those undergoing primary tumor surgery. That is the main reason why we conduct this study. We have changed the presentation of the sentence by replacing it with “However, the prognostic factors of patients undergoing primary tumor resection are still ambiguous” in line 219.
- Point 9: lines 282-283: “lacking other important variables, such as Ki-67, chromogranin A, and neuron-specific enolase [29, 30], which may reduce the prediction accuracy of our nomograms.”. Chromogranin and NSE are two well accepted and validated markers for NETs, the authors should improve this sentence taking it into consideration.
Response 9: Thank you for your professional suggestions. According to the reviewer’s suggestion, we have changed the presentation of the sentence by replacing it with “lacking other important variables, such as Ki-67, chromogranin A, and neuron-specific enolase [29, 30]. Regrettably, these characteristics are not included in the SEER database, thus further studies should incorporate Ki-67, chromogranin A, and neuron specific enolase for analysis.” in lines 287-289.
Special thanks to you for your good comments!